# Comprehensive High-Quality Three-Dimensional Display System Based on a Simplified Light-Field Image Acquisition Method and a Full-Connected Deep Neural Network

**DOI:** 10.3390/s23146245

**Published:** 2023-07-08

**Authors:** Munkh-Uchral Erdenebat, Tuvshinjargal Amgalan, Anar Khuderchuluun, Oh-Seung Nam, Seok-Hee Jeon, Ki-Chul Kwon, Nam Kim

**Affiliations:** 1School of Information and Communication Engineering, Chungbuk National University, Chungbuk 28644, Republic of Korea; uchka@chungbuk.ac.kr (M.-U.E.);; 2Department of Electronics Engineering, Incheon National University, Incheon 22012, Republic of Korea

**Keywords:** simplified light field image acquisition, deep neural network, hash encoding, comprehensive 3D display

## Abstract

We propose a high-quality, three-dimensional display system based on a simplified light field image acquisition method, and a custom-trained full-connected deep neural network is proposed. The ultimate goal of the proposed system is to acquire and reconstruct the light field images with possibly the most elevated quality from the real-world objects in a general environment. A simplified light field image acquisition method acquires the three-dimensional information of natural objects in a simple way, with high-resolution/high-quality like multicamera-based methods. We trained a full-connected deep neural network model to output desired viewpoints of the object with the same quality. The custom-trained instant neural graphics primitives model with hash encoding output the overall desired viewpoints of the object within the acquired viewing angle in the same quality, based on the input perspectives, according to the pixel density of a display device and lens array specifications within the significantly short processing time. Finally, the elemental image array was rendered through the pixel re-arrangement from the entire viewpoints to visualize the entire field-of-view and re-constructed as a high-quality three-dimensional visualization on the integral imaging display. The system was implemented successfully, and the displayed visualizations and corresponding evaluated results confirmed that the proposed system offers a simple and effective way to acquire light field images from real objects with high-resolution and present high-quality three-dimensional visualization on the integral imaging display system.

## 1. Introduction

Three-dimensional (3D) light field image acquisition techniques encode all 3D information, such as parallax, depth, shading, and the texture of real or virtual objects, into two-dimensional (2D) elemental images, using incoherent illumination [1,2]. These methods tend to be reliable, effective, and capable of supporting many types of 3D displays. The key to success is the acquisition of high-quality light field images. There are two main types of light field image acquisition techniques: single- and multicamera-based methods [3,4,5,6,7,8,9,10,11,12,13,14,15,16,17,18,19,20,21,22]. Single-camera methods can be divided into depth camera and lens array types. The depth camera consists of two sensors, i.e., infrared rays and color sensors, such that the 3D information of the real object is detected simultaneously [3,4,5,6]. The lens array-based single-camera method has traditionally been used for light field imaging as it employs a simple configuration in which the lens array is located between the camera lens and the sensor [7,8,9,10,11,12,13,14]. Here, each elemental lens of the lens array provides different viewpoints of the object; the full 3D information of the object is encoded in the disparity between the captured elemental images. By changing the specifications of the elemental lenses, the depth range of image acquisition, viewing angle, and resolution can be controlled. This method is a simple and efficient representative method capable of acquiring light field images. It was successful in reconstructing fine-quality point cloud models with data obtained from real-world objects through lens array-based light field cameras manufactured by Lytro Inc. [12]. Also, a virtual 3D model-free approach for recovering 3D facial geometry based on a single light field camera and a convolutional neural network using Epipolar plane images was introduced [13]. Thus, it was confirmed that such a lens array-based single-camera method is appropriate for light field imaging techniques. However, the resolution of each elemental image tends to be low as it is affected by optical aberrations of the lenses, resulting in poor image quality. Even though resolution and quality improvement methods based on convolutional neural networks have been applied, they are still insufficient for comfortable observation [13,15,16]. Multicamera-based methods, also known as camera array methods, acquire light field images with high resolution and high quality via multiple interconnected digital cameras [17,18,19,20,21,22]. The main configuration can be one-dimensional, i.e., horizontal or vertical parallax only, or 2D, such that it captures full-parallax images with each camera, providing different viewpoints of the object, similar to a lens array. Light field images can also be captured through parallel or toe-in methods, according to the application. Multicamera-type methods are used to obtain high-resolution and high-quality images of real objects from various perspectives; however, the structures tend to be bulky as they must accommodate a certain number of cameras, and the interconnections between cameras are complex. As such, additional processes are required to correct lens distortion and brightness in each camera. Therefore, it is important to implement the high-quality light field image acquisition method with a simple structure.

Recently, several methods of generating high-quality 3D models from given perspectives based on convolutional neural network models have been introduced [15,16,23,24,25,26,27,28]. For example, some methods estimate the depth of information from the perspectives and generate 3D models for the object [23,24], and other methods represent the view synthesis of 2D images as 3D visualizations [25,26,27,28]. Even though these methods can provide much higher quality 3D visualizations than before, it is difficult to reconstruct high-quality natural-view 3D images that meet our demands through these methods.

In this paper, we propose a comprehensive high-quality 3D display system based on a simplified light field image acquisition method, including a simple structure and high-resolution/high-quality image capturing, as with camera array methods, and a custom-trained full-connected deep neural network model. The main goal of the proposed system is to display the final reconstructed 3D image as high-quality as possible; the high-quality data are obtained from the acquisition stage and high-quality view synthesis is regenerated through a custom-trained full-connected deep neural network within a general environment. First, a moving camera array captures the high-resolution perspectives automatically with the toe-in configuration according to the inputted parameters; then, the brightness distribution is corrected with a histogram-matching process based on the average histogram. Since the proposed system requires the elemental image array (EIA) according to the feature of integral imaging display, a custom-trained full-connected deep neural network model is used to generate the overall viewpoints of the object, based on the corrected perspectives. Finally, the EIA is generated through the pixel re-arrangement method from the view synthesis. Optical experiments were performed to verify that the proposed method can serve as a simple and effective way to display natural-view high-quality light field images of real objects.

## 2. Proposed Method

Figure 1 shows a schematic configuration of the proposed high-quality 3D display system based on a simplified light field image acquisition method. The proposed system consists of main three sections: acquisition for high-resolution perspectives, generation of the view synthesis, and EIA generation and 3D display.

### 2.1. Simplified Light Field Image Acquisition Method

Figure 2a shows the simplified light field image acquisition method consisting of a minimized camera array and a multifunctional motor. Here, the camera array consists of *N_c_* high-resolution digital cameras installed and fixed along the vertical axis. When the camera array acquires the light field images with the parallel method, all cameras focus on the object to capture the image. Note that the camera array can capture the perspectives via parallel or toe-in configurations, and a toe-in method is suitable for the proposed system. Here, the central camera focuses on the object as-is, and the other cameras rotate up or down at a given angle based on the central camera’s focusing position. A multifunctional smart motor is mounted on the stage such that the camera array can be shifted horizontally along the stage.

Figure 2b shows a flowchart of the automatic capturing process. The integrated controller supervises the overall process of capturing light field images from the real object. The computer is interconnected with the smart motor controller and camera shutter controller simultaneously and synchronizes the operations of the connected devices. Additionally, to interconnect the cameras and control their shutters simultaneously, we incorporated Raspberry Pi with the switch circuit board. The cameras are connected to the switch circuit board by universal serial bus ports, the switch circuit board is connected to the Raspberry Pi, and the Raspberry Pi is connected to the computer via a Wi-Fi connection using PuTTY, which is an open-source terminal emulator, serial console, and network file transfer application. In the integrated controller, the total number of perspectives (*N_p_*), a step of the X-motor responsible for shifting the interval of the camera array horizontally (*X_i_*), a step of the Y-motor controlling the interval of rotation of the camera array (*θ_i_*), and the initial position of the camera array corresponding to the position of the first perspectives for the X-motor (*P*_0_) and Y-motor (*R*_0_) are inputted by the user, where X-motor and Y-motor indicate the shifting and rotating units, respectively. Note that the integrated controller does not consider the number of cameras *N_c_*. All the user must do is input the main parameters and execute the system operation and the system starts to capture the perspectives automatically. Here, the camera array is shifted and rotated to the *P*_0_ and *R*_0_ positions immediately after the main parameters are inputted, and the integrated controller sends the command to execute a Python script on Raspberry Pi to control the camera shutters; the first series of perspectives are captured and saved as the image files. Then, in each step of *X_i_* and *θ_i_*, the camera array captures the perspectives. When *N_p_*-th perspectives are completely captured, the smart motor and translation stage is stopped, and the camera shutters are deactivated; this disables the Raspberry Pi connection. Here, the cameras with only minimized numbers, *N_c_*, are used (instead of *N_c_* × *N_p_* cameras), but the desired *N_c_* × *N_p_* perspective can be obtained; thus, the proposed system can be considered economically less burdensome. Note that *N_c_* × *N_p_* perspectives are captured and stored immediately in the computer as image files, without using any additional data storage.

Before the generation of view synthesis, the color and brightness distribution of all images should be corrected in an identical manner as a pre-processing. This is because the location of the ambient light source and the shape of the object affect the brightness distribution of the images captured by each camera, i.e., the color of each image is slightly different. In the proposed system, after all the perspectives are captured, the average histogram value of the central perspectives captured by each camera is calculated. The histograms of the other perspectives are matched to the calculated average histogram, as shown in Figure 3. This process facilitates uniformity among the perspectives such that they are smoother and brighter and thus more similar to the reference histogram than the original images. Given that each perspective contains information from different viewpoints, the 3D light field display can be provided with high-quality 3D data in the proposed system.

### 2.2. Train for an Instant-NGP Model with Hash Encoding

Since the number of cameras is minimized, the 3D image cannot be displayed properly because the number of perspectives should be *N_p_* × *N_p_*, according to the 3D display feature. We considered an integral imaging 3D display including a high-resolution display device as the final output of the proposed system whereby intermediate-view image generation methods are utilized to fill the missing viewpoints; however, there is a disadvantage in that the generated image quality is too low, and it becomes worse when the process is iterated twice or more [1]. Therefore, in the proposed system, in order to display a high-quality 3D image properly, an instant neural graphics primitive (instant-NGP) model, a lightweight simple full-connected deep neural network [28], was trained to extract the high-quality view synthesis of the object in order to match *N_p_* × *N_p_* perspectives. Compared to the existing neural radiance fields (NeRF) method [26,27], the basic configuration and operation of the model are similar, but it has the advantage of being able to extract higher quality perspectives in a shorter processing time because the instant-NGP model creates a small neural network by performing the multiresolution hash encoding with the parallelized architecture of graphics processing units (GPUs). Note that the conventional NeRF method uses positional encoding to represent a high-frequency function and samples from the high-frequency function using a hierarchical sampling procedure to form a multilayer perceptron. However, the high-frequency function is computationally complicated, and frequency collisions often occur; thus, using a double hierarchical sampling procedure is very slow for multilayer perceptron training.

Figure 4 shows the illustration of the view synthesis generation of the object through the custom-trained instant-NGP model. The general procedure of the model is generating a high-quality 3D scene including all viewpoints of the object from the input perspectives and rendering the high-quality 2D perspectives. As mentioned before, an instant-NGP uses a small network through multiresolution hash encoding with GPU parallelized architecture. In the hash encoding, first, a spaced grid of input perspectives is created from wide to fine at levels from 2^1^ up to 2^4^ (*L*). The number of *L* defines how many varieties of grids are created where *l* indicates the index of level. The sizes of the widest and finest grids are determined as *S*_min_ and *S*_max_ respectively, and each grid has a different value of corner indices. From Figure 4, the corner indices of the widest and finest grids are illustrated as (*c_w_*_1_, *c_w_*_2_, *c_w_*_3_, *c_w_*_4_) and (*c_f_*_1_, *c_f_*_2_, *c_f_*_3_, *c_f_*_4_), and the average values of corner indices in each grid cannot be equal to other grids’ values. This process gives an opportunity to eliminate hash collisions and the basis of more accurate calculations. Then, the corner indices that are the corresponding hashed integer coordinates are stored at the corresponding hash tables (lookup tables), and they are linearly interpolated according to the relative position in the input 3D scene. Note that a linear interpolation method is a curve fitting using linear polynomials to construct new viewpoints within the range of a discrete set of overall viewing zones, and linear interpolation-based intermediate-view image generation methods are widely used in light field imaging techniques [1,2]. Finally, the full-connected deep neural network extracts *N_p_* × *N_p_* perspectives according to the camera poses, which are determined by the view parameters of the input perspectives.

Unlike the NeRF, the instant-NGP model uses an Adam optimizer in the training, which is implemented by the tiny-cuda-nn framework [28]. According to this framework, we trained the instant-NGP model to extract the desired view synthesis from the input perspectives through the following processing steps, as shown in Figure 5. Here, first, the model determines the camera poses of input perspectives according to the pre-defined view parameters. From the camera poses of input perspectives, the grids are generated with *L* levels, and hashes are encoded with linear interpolation. Then, the multilayer perceptron proceeds according to the encoded hashes and defines the camera poses of *N_p_* × *N_p_* cameras. Simply put, the multilayer perceptron is a deep neural network that expects the 3D volumetric scene of the original object, and a lot of iterations are performed in order to obtain a good-quality multilayer perceptron. Note that the camera pose is a ray tracing calculation in the virtual environment that is performed for each image. Since the instant-NGP uses a volumetric rendering method, an imaginary 3D volume of the object is depicted via the COLMAP structure from the motion package, derived from the camera poses of input images in the virtual space. When the *N_p_* × *N_p_* camera pose was created, we set the central row as 0° vertically; the central camera of the central row was set to look straight to an imaginary 3D volume, and the interval between each neighboring camera in the same row was set as the *θ_i_* interval. Also, the interval between each neighboring row was set as the same value. Finally, in the multilayer perceptron, we needed to check whether the top, bottom, and center rows of the *N_p_* × *N_p_* camera poses matched the input camera poses and whether the related perspectives were rendered. Since the instant-NGP model can be performed 100 times faster than the conventional NeRF model and is trained with the designed scheme, it can obtain a higher-quality view synthesis with less loss of quality in the proposed system. Note that the training parameters should consider the EIA specifications of the next stage.

### 2.3. Pixel Re-Arrangement Method-Based EIA Generation

Finally, the EIA is generated from the rendered perspectives and displayed on the display unit. Here, the EIA is generated by the pixel re-arrangement method to visualize the entire encoded field-of-view for which each pixel of EIA is re-arranged from the newly rendered perspectives [29,30]. First, the main parameters including the number of virtual cameras, the distance between neighboring cameras, and the distance between the camera and image planes are optimized. Note that the field-of-view of the virtual camera array is set as the acquired viewing angle of the object. Then, each elemental image with the (*i*,*j*) index, *EI_ij_*, can be calculated as follows:(1)$$EI_{ij}\left(s,t\right)=P_{st}\left(i,j\right);\;s=1,\ldots ,N_{p},\;t=1,\ldots ,N_{p},\;i=1,\ldots ,m,\;j=1,\ldots ,n,$$
where *P_st_* is the set-of perspective images, *s* and *t* denote the position of the *N_p_* × *N_p_* virtual cameras along horizontal and vertical axes, respectively, and *m* and *n* are the numbers of pixels of each virtual perspective in horizontal and vertical directions, respectively. Here, the pixel number that belongs to the single elemental lens is the most important parameter in the EIA generation. Accordingly, the entire EIA is generated and displayed on the high-resolution display device and directly reconstructed as a natural-view 3D image when the lens array is mounted in front of the display device. This kind of EIA generation keeps the overall acquired field-of-view of the object on the 3D display.

## 3. Experiment and Results

The overall stages and related processes of the proposed system were designed and analyzed, and the following experiments and measurements were conducted by the authors to confirm the characteristics and superiority of the proposed system.

### 3.1. Simplified Light Field Image Acquisition Performance

Figure 6 shows the prototype of the proposed light field image acquisition system. The experimental setup consisted of a computer with an Intel Core i7-12700KF (3.60 GHz) central processing unit, 32 GB of memory, an NVIDIA GeForce RTX 2070 GPU, and a semiconductor-based storage device with 512 GB capacity; the camera array comprised three Sony α6000 high-resolution digital cameras (*N_c_* = 3; resolution: 6000 × 4000 pixels) and a multifunctional smart motor with a Suruga Seiki DS102 controller on a translation stage 600 mm in length. The minimum shifting distance of the X-motor was 0.01 mm (the maximum distance depended on the length of the translation stage, which was 600 mm in the experiment) and the minimum rotating angle of the Y-motor was 0.004° (the maximum was unlimited, i.e., 360°). With this setup, it was possible to capture a sufficiently large number of perspectives within the length range of the translation stage. To interconnect the cameras and control the shutters simultaneously, Raspberry Pi3 was incorporated into the switch circuit board via general-purpose input/output pin connections.

Figure 7a shows the user interface of the integrated controller. The controller was built on the basis of LabVIEW software and synchronized with the operation of the smart motor and camera shutters. When the “Capture” button was pressed after inputting the main parameters, the motor shifted and rotated the camera array accordingly, and the camera array captured the perspectives. Here, the Raspberry Pi3 board controlled the camera shutter through a Python script to capture all perspectives simultaneously. An integrated controller applied a delay time to reduce the effect of vibration on the camera array during image acquisition; in our experiments, the delay time was set to 1 s. Figure 7b shows the brightness distribution difference between the originally captured and corrected perspectives. In the calculation of average brightness, only the background was captured, and we could verify that the brightness was increased from the bottom to the top camera. However, after the histogram matching process, the brightness distribution of all images became identical.

In the experiment, the camera array was shifted horizontally by 597.8 mm and the horizontal viewing angle of the acquired light field image was 20.544°. To match the vertical and horizontal viewing angles, the distance between the neighboring cameras was set to approximately 180 mm such that the vertical viewing angle of the acquired light field image was set as approximately 20.4° with the toe-in capturing method. The distance between the camera and the object was approximately 1000 mm and the number of perspectives along the horizontal axis was *N_p_* = 11, where the interval between neighboring perspectives was 59.78 mm and 2.0544°. Note that *N_p_* is not limited in the proposed system, given that a total of 3 × 60,000 perspectives can be captured using the translation stage with a 600 mm length. Figure 8 shows an example of 3 × 11 perspectives for a real object, “Belle”, with a height of approximately 1 m. Cam1, Cam2, and Cam3 indicate the camera numbers from bottom to top; 1, 6, and 11 are the horizontal indices.

The resolution of each perspective was 6000 × 4000 pixels, i.e., the same as the camera resolution, where the total imageset (3 × 11 perspectives) size was approximately 136.6 MB in a computer that did not require additional data storage. Therefore, the simplified light field image acquisition system can acquire light field images with a much improved resolution and a larger viewing angle compared to conventional lens array-based commercial light field cameras or depth cameras. As shown in Table 1, only three cameras were utilized to obtain 3 × 11 perspectives, giving the same task to 33 cameras of the camera array; concerning the advantages of the system, it can be observed that the cost performance was good.

### 3.2. Instant-NGP Model Training and Generation for View Synthesis

Figure 9 shows the user interface of the training process for an instant-NGP model and generation for view synthesis. Both procedures were performed on the Python script. In the training, the iteration of the deep neural network was set as 2122 times within the 30 s and the detected loss of training was 0.00195. In the view synthesis generation, by using the custom-trained instant-NGP model, 11 × 11 camera poses were determined within approximately 4.04 s (30 fps) from the input 3 × 11 perspectives and 11 × 11 perspectives were rendered with a resolution of 4000 × 4000 pixels within approximately 20 min. Since each element lens of the 3D display unit was shaped as a square, the resolution of each rendered perspective was set to 4000 × 4000 pixels, where the overall size in the computer was approximately 339 MB. Unlike previous similar methods [12], the view synthesis generation process becomes simpler by filling in the rest of the viewpoints in consideration of the parameters used to generate the 3D model from the input data, rather than regenerating the 3D model. Note that the output of a custom-trained instant-NGP model is not limited. It can render more images, for example, 50 × 50 or 500 × 500 and more; it only depends on the number of pixels in each elemental image.

Figure 10 shows an example of the newly rendered 11 × 11 perspectives. The number after lines presents the indices of perspectives in each row. Note that the conventional NeRF model requires approximately 1–2 days to output fine-quality images [26]; therefore, a custom-trained instant-NGP model has the significant advantage of being able to produce sufficiently high-quality perspectives in a very short time by performing multiresolution hash encoding.

For the quality and accuracy measurements of the rendered perspectives, we compared the input and output perspectives with the same viewpoint index using peak signal-to-noise ratio (PSNR) and structural similarity index measure (SSIM) methods. Here, PSNR and SSIM values were 27.33 dB and 0.88, respectively, where the corresponding input central perspective (Cam2-6 of Figure 8) was utilized as a reference, as shown in Figure 11. Therefore, it could be observed that the loss of output images of the custom-trained instant-NGP model was significantly less, and both images were almost identical.

### 3.3. EIA Generation and 3D Display

From the rendered set of different viewpoints, the EIA was generated for the display unit through the pixel re-arrangement process. Here, the field-of-view was given at 20.544° (horizontal)/20.4° (vertical), which was identical to the acquired fields-of-view. An integral imaging display consisted of a 15″ display device with 4K resolution (3840 × 2160 px) and a lens array with 345 × 194 lenses (1 mm pitch and 3.3 focal lengths), as shown in Figure 12a. The EIA with 194 × 194 elemental images was generated where the overall resolution was 2160 × 2160 px and each elemental image consisted of 11 × 11 px. In the experiment, two EIAs were generated and reconstructed for the entire object and for the upper part of the object for more detailed observation, as shown in Figure 12b.

Figure 13 shows optically reconstructed 3D images on the integral imaging display from two EIAs. Here, we verified that the proposed system successfully encoded 3D information about an object, and the virtual 3D images were reconstructed clearly, approximately 20 mm backward from the display device, with perfect depth cues. The viewing angle of the 3D display was approximately 13° according to the lens array’s specifications; however, approximately 20° of fields-of-views of the object were visualized within the 3D display’s viewing angle. Note that the proposed system was not designed only for human faces, but it was confirmed that facial details could be expressed well in the proposed system, like with existing 3D face recovery methods [13].

Although the quality of the reconstructed 3D images was good, we evaluated the quality through quantitative evaluation methods in order to try to prove the quality clearly. Here, the non-reference-type naturalness image quality evaluator (NIQE) [32] and perception-based image quality evaluator (PIQE) [33] methods were used to measure the quality of the displayed images. Note that the reconstructed 3D images were captured manually. Therefore, the PSNR-like reference-type evaluation methods could not be utilized by using the acquired or rendered perspectives as a reference; it was appropriate to separately evaluate the corresponding image itself. Table 2 shows the evaluated NIQE and PIQE scores for the central viewpoints of the reconstructed 3D images that were presented in Figure 13a,b. For the PIQE, the images were evaluated through the default model; for the NIQE, the scores were measured through the custom models that were trained for each image in order evaluate more clearly. These results prove that even though the 3D image was reconstructed through a lens array, various optical issues such as lens aberrations could occur, but the proposed system displayed 3D images of significantly high quality. Accordingly, it confirmed that the proposed system offers a simple and effective way to acquire high-quality and high-resolution light field images of real objects and display high-quality 3D visualizations within the ordinary environment. Note that the values of both methods are inversely correlated to the perceptual quality of images; therefore, a low value indicates better quality [32,33].

## 4. Conclusions

A high-quality light field 3D display system based on a simplified light field image acquisition method and a custom-trained deep neural network was proposed and implemented. The existing research related to commercial light field cameras, such as 3D model reconstruction or 3D face recovery methods, has been presented successfully, but the purpose of the proposed system is slightly different from these methods. The ultimate goal of the proposed system is to design/implement a high-quality 3D display system, including a high-quality and high-resolution 3D data acquisition method from real-world objects within an ordinary environment and an integral imaging 3D display unit based on a high-resolution display device and large-sized lens array. The proposed system consists of main three stages, and we designed and implemented each stage and combination. The simplified light field image acquisition method acquired high-resolution and high-quality 3D information of a real object (3 × 11 images) through only three high-resolution cameras with a dual-function smart motor that could originally be acquired through a camera array consisting of numerous (33) cameras. In order to match the viewpoints of the real object and the configuration of the 3D display unit, a full-connected instant-NGP deep neural network with a multiresolution hash encoding model was utilized that could render the view synthesis of the object with higher quality within a much shorter processing time compared with the other methods such as NeRF and COLMAP networks. In the experiment, an instant-NGP model was trained to extract 11 × 11 perspectives from acquired images with the same quality. Finally, EIAs were generated from the rendered perspectives through a pixel re-arrangement process and displayed/reconstructed as high-quality 3D images. The experimental results confirm that the proposed system provides a simple and effective way to acquire high-resolution and high-quality light field images of real objects and provide high-quality natural-view 3D images with accurate 3D information. In particular, it was certified that high-quality light field image acquisition/display and image processing can be performed at a relatively high speed in an ordinary environment, without any special equipment and at a low cost. Future research should focus on the fast computation of the proposed system.

## Figures and Tables

**Figure 1 sensors-23-06245-f001:**
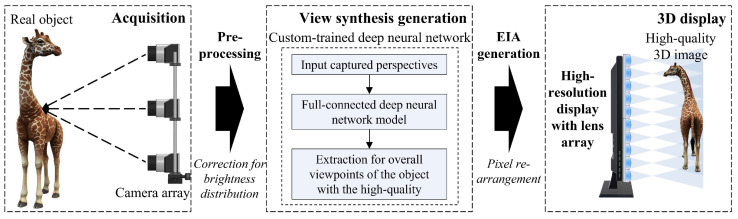
Overall schematic configuration of the proposed high-quality 3D integral imaging display system based on the simplified light field image acquisition method.

**Figure 2 sensors-23-06245-f002:**
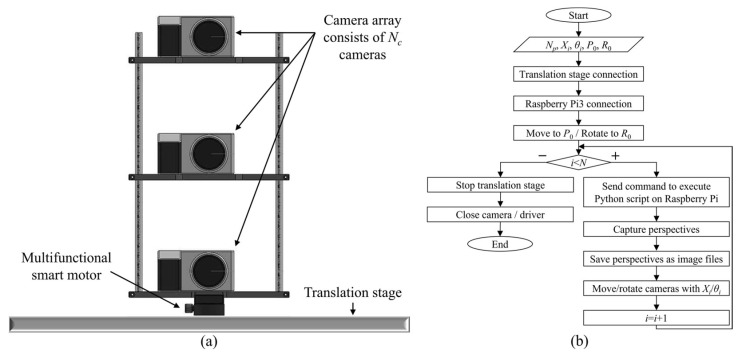
(**a**) The structure of the simplified light field image acquisition system and (**b**) a flowchart of the main procedure of the integrated controller.

**Figure 3 sensors-23-06245-f003:**
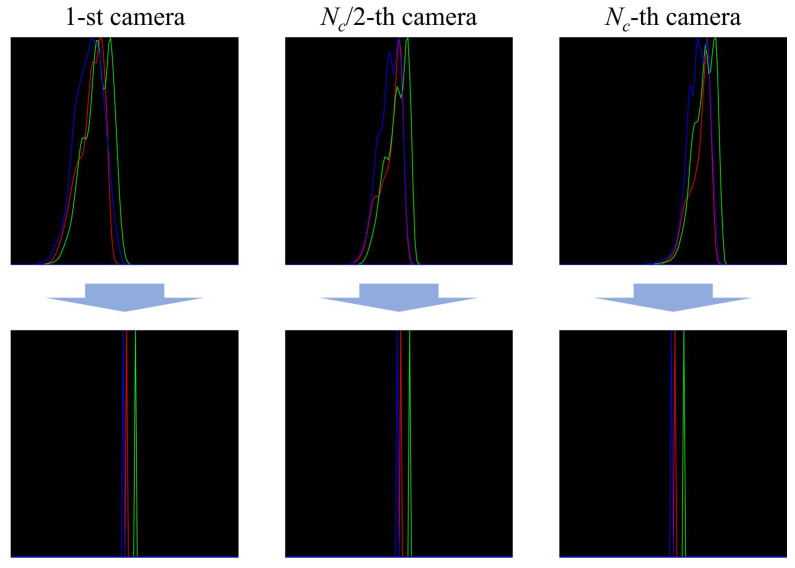
Histogram matching for the perspectives captured by different cameras using a calculated average histogram.

**Figure 4 sensors-23-06245-f004:**
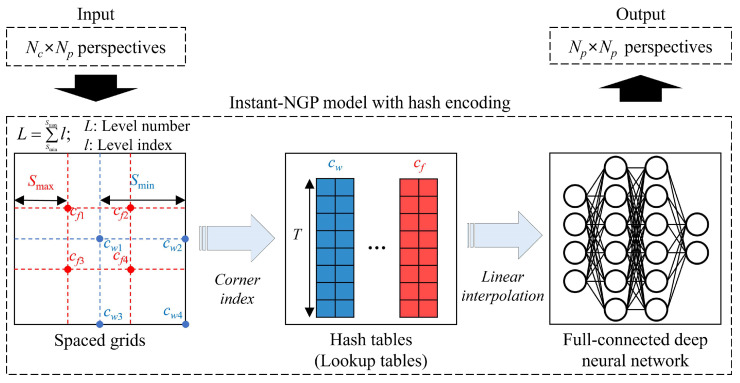
Basic configuration for the rendering of *N_p_* × *N_p_* perspectives from input *N_c_* × *N_p_*-corrected perspectives through the custom-trained instant-NGP model.

**Figure 5 sensors-23-06245-f005:**
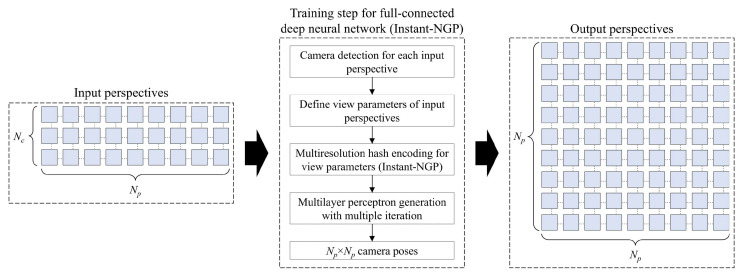
The training scheme for the instant-NGP model to extract *N_p_* × *N_p_* perspectives.

**Figure 6 sensors-23-06245-f006:**
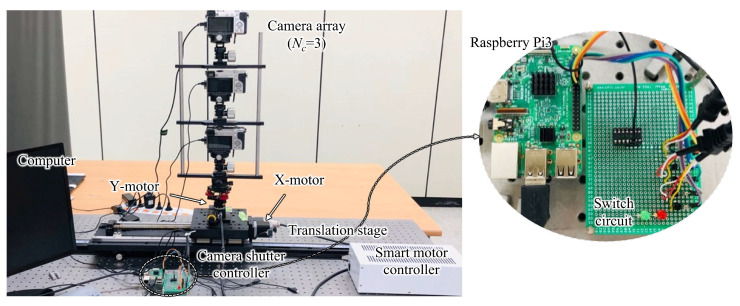
Prototype of the simplified light field image acquisition system of the proposed system (Appendix A: The performance of a simplified light image acquisition system).

**Figure 7 sensors-23-06245-f007:**
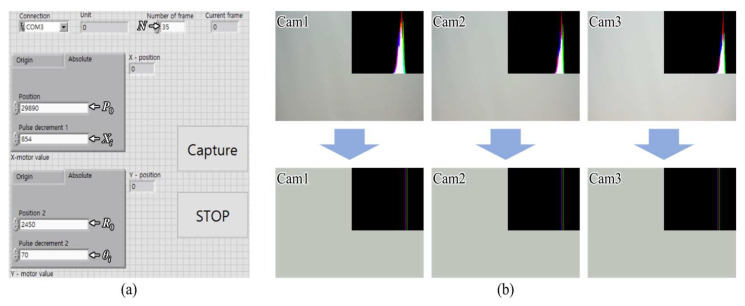
(**a**) User interface of the computer-based integrated controller running on LabVIEW software and (**b**) the brightness distribution for each camera before (**top**) and after (**bottom**) the histogram matching process.

**Figure 8 sensors-23-06245-f008:**
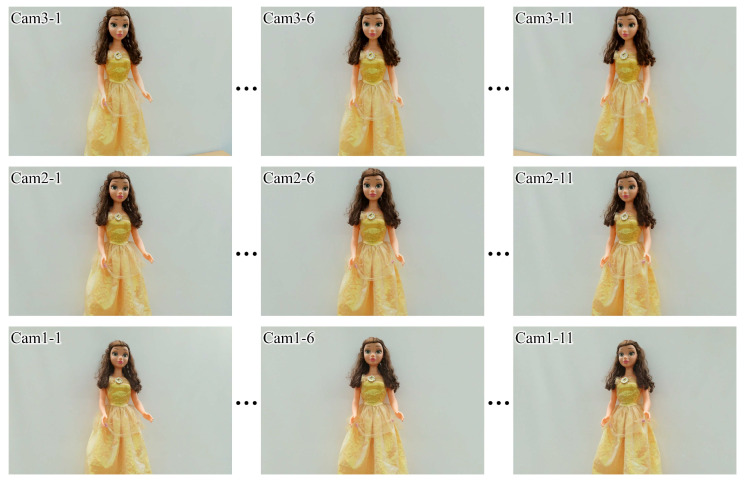
The 1st, 6th, and 11th captured perspectives of a real object, “Belle”, for each camera after brightness distribution correction.

**Figure 9 sensors-23-06245-f009:**
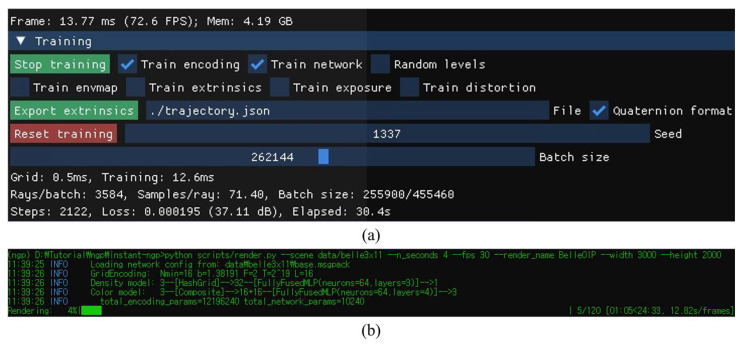
The user interface of (**a**) the training for an instant-NGP model and (**b**) the rendering of 11 × 11 perspectives according to the calculation of camera poses.

**Figure 10 sensors-23-06245-f010:**
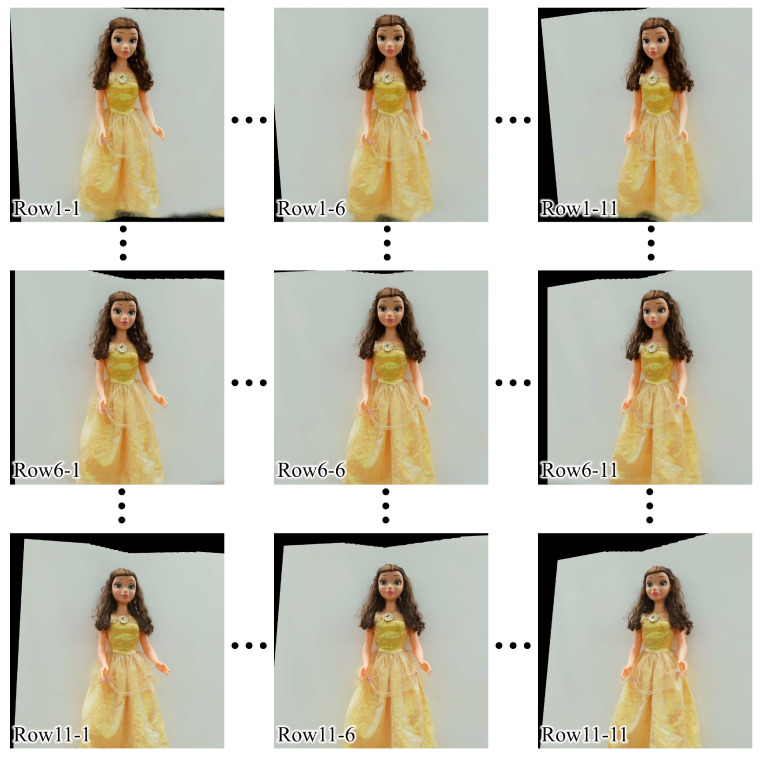
The 1st (**left-most**), 6th (**central**), and 11th (**right-most**) perspectives of the 1st (**top**), 6th (**central**), and 11th (**bottom**) rows of 11 × 11 perspectives.

**Figure 11 sensors-23-06245-f011:**
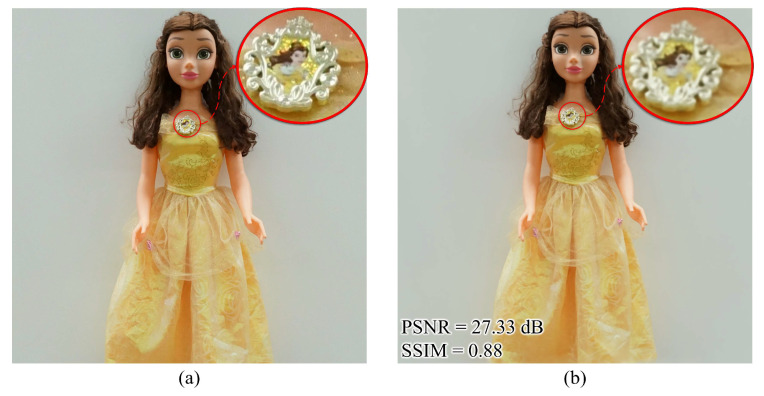
The comparison between the central (6th) perspectives of (**a**) captured/corrected and (**b**) newly rendered 11 × 11 images.

**Figure 12 sensors-23-06245-f012:**
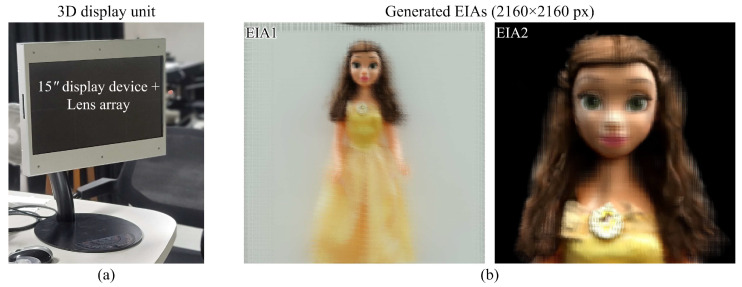
(**a**) A prototype of the integral imaging 3D display and (**b**) the EIAs for 3D display generated from the entire object (**left**) and upper part (**right**).

**Figure 13 sensors-23-06245-f013:**
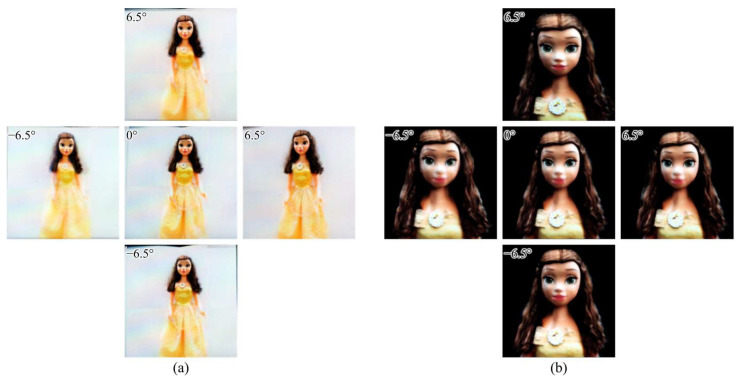
Reconstructed 3D image of the object from various viewpoints: center (0°), left (−6.5°), right (6.5°), top (6.5°), and bottom (−6.5°) views (**a**) for the entire object (Appendix A: The 3D image reconstruction for entire object) and (**b**) for the upper part of the object (Appendix A: The 3D image reconstruction for the upper part of the object) in order to present more detailed observation.

**Table 1 sensors-23-06245-t001:** Comparison for 3 × 11 perspective acquisition based on a commercial light field camera, the camera array, and the proposed system.

Characteristics	Commercial Light Field Camera [31]	Camera Array (3 × 11 Cameras)	Proposed System (3 Cameras)
Resolution (single perspective)	625 × 434 px	6000 × 4000 px	6000 × 4000 px
Data quality (for proposed system)	Low	High	High
Data quantity	Common	Common	Common
Data storage	NA	Can be required	NA
Viewing angle	Narrow (fixed)	Wide (can be controllable)	Wide (can be controllable)
Depth-of-field	Common (fixed)	Wide (can be controllable)	Wide (can be controllable)

**Table 2 sensors-23-06245-t002:** Measured NIQE and PIQE scores for the central viewpoints of reconstructed 3D images.

Evaluation Method	Central Viewpoint (0°) of 3D Image of Figure 13a	Central Viewpoint (0°) of 3D Image of Figure 13b
NIQE	3.31	3.19
PIQE	20.06	18.57

## Data Availability

Not applicable.

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
