# Peer review of "Comprehensive High-Quality Three-Dimensional Display System Based on a Simplified Light-Field Image Acquisition Method and a Full-Connected Deep Neural Network"

_sensors, 2023, doi:10.3390/s23146245_

Round 1

Reviewer 1 Report

Good works. Can be accecpted in the present form.

Author Response

The authors sincerely appreciate the reviewer for taking the time to review the manuscript. Their compliment comment has encouraged the authors’ future research.

Reviewer 2 Report

The authors present a method of 3D reconstruction from light field image system designed by them. The authors train a fully connected deep learning framework to output views with consistent quality. Also, the image array from multiple view points is synthesized into a 3D object.

There has been quite some work in this field and it is surprising that the authors have not compared to commercially available light field cameras.  In terms of 3D construction, there has been work on analytically creating 3D reconstructions as well as learning based work (Perra, C., Murgia, F., & Giusto, D. (2016, December). An analysis of 3D point cloud reconstruction from light field images. In 2016 Sixth International Conference on Image Processing Theory, Tools and Applications (IPTA) (pp. 1-6). IEEE., Feng, M., Gilani, S. Z., Wang, Y., & Mian, A. (2018). 3D face reconstruction from light field images: A model-free approach. In Proceedings of the European conference on computer vision (ECCV) (pp. 501-518).) There needs to be a detailed comparison in terms of results with the previous work as well.

.

Reviewer 3 Report

I commend the authors for their fine work. In my view the experimental part has been conducted with care, and sufficient details are presented for the work to be reproduced. In my opinnion this article can be accepted in its current form.

Author Response

(The authors gave the same response as above.)

Round 2

Reviewer 2 Report

The authors have presented an argument that the commercial light field cameras having fixed specifications would lead to degradation of the final reconstructed 3D image. However, it might be important to do an analysis (using simulations) as to the cost vs benefits of having really high-quality reconstructed 3D image in terms of data quality, data quantity, cost of data storage for large data collections etc.
